# A New Cell Model for Investigating Prion Strain Selection and Adaptation

**DOI:** 10.3390/v11100888

**Published:** 2019-09-22

**Authors:** Alexandra Philiastides, Juan Manuel Ribes, Daniel Chun-Mun Yip, Christian Schmidt, Iryna Benilova, Peter-Christian Klöhn

**Affiliations:** MRC Prion Unit at UCL, UCL Institute of Prion Diseases, 33 Cleveland Street, London W1W7FF, UK; a.philiastides@prion.ucl.ac.uk (A.P.); j.ribes@prion.ucl.ac.uk (J.M.R.); d.yip@synthace.com (D.C.-M.Y.); c.schmidt@prion.ucl.ac.uk (C.S.); I.Benilova@prion.ucl.ac.uk (I.B.)

**Keywords:** prions, prion strains, neurodegeneration, selective neuronal vulnerability, adaptation, prion propagation, astrocytes, neurons

## Abstract

Prion diseases are fatal neurodegenerative diseases that affect humans and animals. Prion strains, conformational variants of misfolded prion proteins, are associated with distinct clinical and pathological phenotypes. Host-strain interactions result in the selective damage of distinct brain areas and they are responsible for strain selection and/or adaptation, but the underlying molecular mechanisms are unknown. Prion strains can be distinguished by their cell tropism *in vivo* and *in vitro*, which suggests that susceptibility to distinct prion strains is determined by cellular factors. The neuroblastoma cell line PK1 is refractory to the prion strain Me7, but highly susceptible to RML. We challenged a large number of clonal PK1 lines with Me7 and successfully selected highly Me7-susceptible subclones (PME) to investigate whether the prion strain repertoire of PK1 can be expanded. Notably, the Me7-infected PME clones were more protease-resistant when compared to RML-infected PME clones, which suggested that cell-adapted Me7 and RML are distinct prion strains. Strikingly, Me7-refractory cells, including PK1 and astrocytes in cortico-hippocampal cultures, are highly susceptible to prions, being derived from homogenates of Me7-infected PME cells, suggesting that the passage of Me7 in PME cells leads to an extended host range. Thus, PME clones represent a compelling cell model for strain selection and adaptation.

## 1. Introduction

Prion diseases or transmissible spongiform encephalopathies (TSEs) are transmissible, invariably fatal neurodegenerative diseases that affect animals and humans and include scrapie in sheep, bovine spongiform encephalopathy in cattle, and Creutzfeldt-Jakob disease (CJD), Gerstmann-Sträussler-Scheinker syndrome (GSS), and fatal familial insomnia (FFI) in humans. TSEs are typified by extended periods of asymptomatic disease, followed by clinical disease onset with rapid progression and death within months. Prions, the infectious agents of prion diseases, are thought to arise by template-assisted conversion of the cellular prion protein [1,2], a glycosylphosphatidylinositol (GPI)-anchored membrane protein that is highly expressed in the central nervous system (CNS). Typical pathological hallmarks of prion disease are neuronal loss, spongiform degeneration, gliosis, and prion protein aggregation [3].

Neurodegenerative diseases are characterized by a progressive loss of neurons, but at early stages, pathology only affects particular neurons, a phenomenon that is referred to as selective neuronal vulnerability. The selective loss of vulnerable neurons in early Alzheimer’s disease affects neurons of the entorhinal cortex, the hippocampus, the basal forebrain, and the locus coeruleus [4]. The clinical symptoms of Parkinson’s disease have been associated with a loss of dopaminergic neurons in the substantia nigra pars compacta, corpus striatum, and brain cortex [5,6]. In amyotrophic lateral sclerosis, upper motor neurons in the motor cortex and motor neurons in the spinal cord and in the brainstem are selectively vulnerable. FFI is associated with severe neuronal loss in the thalamus [7] and marginal tissue pathology in the cortex [8]. In other prion diseases, including sporadic CJD (sCJD), GSS, variant CJD (vCJD), and Kuru, degeneration of GABAergic inhibitory interneurons has been reported [9].

Prion strains, conformational variants of misfolded prion proteins, are associated with distinct clinical and pathological phenotypes. Prion strains are typically distinguished by their differences in disease incubation time, lesion profiles, and electrophoretic mobilities, but more recently evidence for strain-dependent differences in cell tropism has been found [10,11,12,13,14]. An immunohistological study provided *in vivo* evidence that the mouse-adapted prion strains Me7 and 22L were preferentially associated with neurons and astroglia, respectively [13], which supports the notion of strain-dependent differences in cell tropism. Such differences have been previously reported in tissue culture. The cell panel assay [12] utilizes immortalized cell lines with distinct susceptibilities to the mouse-adapted prion strains and allows the discrimination of prion strains within two weeks. A role of *Prnp* polymorphism was excluded, since all the cell lines express the Prnp^a^ allele [12]. That prion strains may interact with different protein targets has been inferred from strain-dependent differences in the cellular uptake routes of the prion strains 22L and RML [14]. While differences in the cell tropism of distinct prion strains suggests a contribution of yet unknown cellular factors, the cell panel assay does not lend itself to identifying such factors, since the cells were derived from different tissues or cancers and thus differ substantially in their gene and protein expression profiles. We recently showed in the scrapie cell model N2aPK1 (PK1) that the close genetic kinship between cognate (isogenic) prion-susceptible and -resistant cells, derived by single cell cloning, can be exploited to identify the genes that are associated with prion replication [15].

In the present study, we examined whether the prion strain repertoire of PK1 cells can be expanded to establish isogenic cell clones with distinct susceptibilities to prion strains. While inherently susceptible to RML [15,16] and 22L [12], PK1 is refractory to Me7 [16]. We here report an unexpected progressive enrichment of Me7-susceptible subclones (PME2) by serial rounds of subcloning. We concomitantly monitored the changes in susceptibility of PME2 clones to 22L and RML. Notably, the cell-adapted Me7 prions showed changes in strain properties on Western blot, when compared to those of Me7, but they retained a higher resistance to PK digestion, when compared to cell-adapted RML prions. Strikingly, Me7-refractory PK1 cells were found to be highly susceptible when infected with prions, being derived from homogenates of chronically Me7-infected PME2 cells, which suggests that a single passage in PME2 cells expanded the host range of Me7 prions. We further assessed whether cell- and brain-adapted prion strains infect primary neuronal cells and report rod-like aggregates of disease-associated PrP (PrP^d^) in astrocytes. Our study provides the first evidence for prion strain adaptation in genetically similar cell clones and brings forth a cell-based tool to investigate the molecular basis of cell tropism. Isogenic cell models with differences in the cell tropism for prion strains might facilitate the quest for strain-dependent gene expression patterns and help to identify protein binding partners of prion strains.

## 2. Materials and Methods 

### 2.1. Cell Lines and Tissue Culture

The mouse neuroblastoma cell line PK1 was derived from Neuro2a cells, as specified previously [15,17]. CAD5, a catecholaminergic cell line and LD9, a murine fibroblast cell line were kindly provided by Sukhi Mahal (Scripps, Florida, USA) and they were derived, as described previously [12]. Unless otherwise specified, the cell lines were cultured in Opti-MEM (Thermo Fisher Scientific, Loughborough, UK), supplemented with 10 % heat-inactivated fetal bovine serum (FBS, Invitrogen) and 1 % penicillin/streptomycin (OFCS). The CAD5 cells were cultured in Opti-MEM, supplemented with 10 % HyClone Bovine Growth Serum (BGS, GE Healthcare Life Sciences, Buckinghamshire, UK) and 1 % penicillin/streptomycin (OBGS). LD9 cells were cultured in Minimum Essential Medium Eagle (MEME, Sigma, Dorset, UK), supplemented with 10 % FBS and 1 % penicillin/streptomycin (PS).

### 2.2. Primary Neuronal Cultures

All of the procedures involving animals were performed under a license granted by the UK Home Office and they conformed to the University College London institutional and ARRIVE guidelines. Unless stated otherwise, all the cell culture reagents were purchased from Thermo Fisher Scientific. Twenty four hours prior to dissection, Nunc Lab-Tek chambered cover glass slides were coated with 1 mg poly-L-lysine (Sigma) per ml of 100 mM borate buffer (pH 8.5), then washed twice with sterile water. Primary cortico-hippocampal cultures were prepared from embryonic e17 FVB mouse brains. Briefly, the cortices and hippocampi were dissected in Ca^2+^/M^g2+^-free Hank’s balanced salt solution (HBSS), supplemented with 10 mM 4-(2-hydroxyethyl)-1-piperazineethanesulfonic acid (HEPES), 2 mM L-glutamine, and 1 % PS (dissection medium). The dissected tissue was enzymatically digested for 15 min. at 37 °C with 0.25% (*m*/*v*) trypsin and 1000 U of benzonase (VWR), washed three times with dissection medium, resuspended in plating medium (Dulbecco’s Modified Eagle’s medium (DMEM), supplemented with 10% (*v*/*v*) heat-inactivated horse serum and 20 U/mL of PS) and triturated with fire-polished glass Pasteur pipettes. Cells were filtered through 70 μm cell strainers, plated out into chambered slides, filled with pre-warmed medium at a cell density of 92,000 cells/cm^2^, and then incubated for 1.5 h in a CO_2_ (37 °C, 5% CO_2_) incubator. The medium was then changed to complete Neurobasal medium, supplemented with 2% (*v*/*v*) B27, 0.25% (*v*/*v*) GlutaMAX and 1% PS. Two days after plating, the cells were infected with a 10^−4^ dilution of brain homogenates, or with 10^−2^ dilution of ribolyzed cell homogenates. The primary neuronal cells were cultured for three weeks, while half of the conditioned medium was replaced with fresh Neurobasal medium twice a week.

### 2.3. Abbreviation of Clonal Cell Lines

Isogenic cell clones, derived from the parental clone PME2, were abbreviated with a unique specifier (number of 96-well plate and position) as hyphenated suffix (for example PME2-6D8). The prion strain in prion-infected PME2 cells was specified in box brackets, for instance PME2 [Me7], in line with the nomenclature by Li et al. [18].

#### 2.3.1. Quantification of Prion Infection and Rates of Prion Replication

We utilized the Scrapie Cell assay (SCA), a quantitative cell-based assay, to determine the number of prion-infected cells essentially as described previously [17]. Differences in the rates of prion propagation (Figure 2C), expressed as the number of PrP^Sc^-positive cells per day, were assessed over the course of two to three cell passages after infection while using SCA.

#### 2.3.2. Quantification of Prion Titers Using the Scrapie Cell Assay in Endpoint Format (SCEPA)

In the SCEPA, 5 × 10^4^ PME2 cells per ml of complete medium are plated out into 24 wells of a 96-well plate per sample. Sixteen hours later, the cells are infected with serially 1:10 diluted brain or cell homogenates. Cells were serially passaged essentially as described previously [17]. The distribution of infected wells is computed from the ratio of negative to total wells ($P\left(0\right)=\frac{neg}{tot})$, where $P\left(0\right)=e^{-m}$ and *m* is the mean number of infectious units per well or volume.

### 2.4. Single Cell Cloning and Cryopreservation

The cells were plated into 10 cm petri dishes at limiting dilution (150–300 cells/plate) and then cultured for 10–12 days. Colonies of monoclonal cells were carefully triturated under microscopic control while using a filtered tip of a P200 pipette and transferred into a well of a 96-well plate, supplemented with 200 µL medium. We split the cells for two to three passages prior to infection, a procedure that synchronizes the number of cells per well, due to variations in the number of cells transferred per individual subclone. To maintain clonal differences in susceptibility to prion strains, the cells were frozen at the earliest time possible. The cells were frozen in 96-well plates prior to infection or in 1.5 mL cryovials, following the first SCA after infection. For cryopreservation, the cell pellets were suspended in OFCS, supplemented with 6 % DMSO (cryovials) and 8 % DMSO (96-well plates) and stored at −80 °C overnight, before transfer into the vapor phase of a liquid nitrogen tank for long-term storage.

### 2.5. Preparation of Cell Homogenates

To prepare cell homogenates from chronically infected and uninfected clones, the cells were grown in 15 cm petri dishes to 80–90% confluency. Cells were resuspended in serum-free medium and then centrifuged for 5 min. at 500× *g*. Cell pellets were then resuspended in 1 mL of serum-free medium, supplemented with protease inhibitor cocktail mix (Merck Millipore, Watford, UK), 4 μL benzonase (25–29 U/µL, Merck Millipore), and zirconium beads. The cells were homogenized with two 60-s cycles at 6500 rpm while using a ribolyzer (Precellys 24, Bertin Instruments, Basingstoke, UK). The homogenates were stored at −80 °C until further processing.

### 2.6. Proteinase K Digestion of Cell and Brain Homogenates

#### 2.6.1. Cell Homogenates

The protein concentration of cell homogenates was determined while using bicinchoninic acid (BCA) assay (Thermofisher Scientific) according to the manufacturer’s specification. Aliquots of 50 μg cell homogenates were mixed with equal volumes of RIPA buffer (50 mM Tris, 150 mM sodium chloride, 0.5% sodium deoxycholate, 1% Triton (*v*/*v*), pH 7.4) and incubated on ice for 15 min. Aliquots of Proteinase K (PK) were added to a final concentration of 1–55 µg/mL (see Figure 3) and the volume of the reaction mix was adjusted to 90 μL with PBS. Samples were incubated on a thermomixer at 37 °C while shaking at a speed of 800× rpm. After 30 min., the samples were placed on ice and PK digestion was stopped by adding 10 μL 4-(2-aminoethyl) benzenesulfonyl fluoride hydrochloride (AEBSF, 100 mM). After adding 100 µL 2× SDS sample buffer, containing 4% β-mercapto ethanol, samples were incubated on a heat block at 100 °C for 5 min. If not used for Western blotting immediately, the samples were stored at −80 °C.

#### 2.6.2. Brain Homogenates

RML and Me7 brain homogenates (10% (*w*/*v*)) were diluted 1:10 into uninfected CD-1 brain homogenate, supplemented with 4 μL benzonase per ml and magnesium chloride at a final concentration of 500 µM, and then incubated on a thermomixer at 800 rpm for 10 min. and 37 °C. Brain homogenates (1%; *w*/*v*) were centrifuged at 100× *g* for 1 min. and supernatant aliquots of 30 μL were digested with 40 µg/mL Proteinase K (Merck, Nottingham, UK) for 1 h at 37 °C and 800 rpm. The samples were then centrifuged at 16,000× g for 1 min. and PK blocked by adding AEBSF. Sample preparation for Western blotting was conducted, as described above.

### 2.7. Western Blotting

The proteins were separated under denaturing conditions by Western blotting using pre-cast 16% Tris-glycine polyacrylamide gels (Invitrogen). Gels were electro-blotted onto polyvinylidene difluoride membranes and blocked for 4 hrs in 5% (*w*/*v*) non-fat milk powder in PBS containing 0.05% (*v*/*v*) Tween-20 (PBST). PrP was detected by incubating immune-blots overnight at 4 °C with biotinylated primary anti-PrP antibody ICSM35-B at a 1: 10,000 dilution, followed by incubation with NeutrAvidin-horseradish peroxidase (HRP)-conjugated secondary antibody (1:7000) for 3 hrs at 4 °C. Blots were then incubated with SuperSignal West Pico plus chemiluminescence substrate, according to the manufacturer’s specification. Biomax (Kodak, London, UK) films were exposed for various time points to avoid signal saturation and corresponding intensities were analyzed while using Image J software after subtracting the background values.

### 2.8. Laser-Scanning Microscopy

Primary cortico-hippocampal cells were cultured as specified in “Primary neuronal cultures” above. Prion-infected and uninfected cells were plated into wells of eight-well chambered cover glass slides at a cell density of 5 × 10^4^ cells/mL of culture medium. The cells were cultured for five days with one medium change. The cells were fixed for 12 min. in 3.7% formalin/PBS and washed with PBS. The fixed cells were incubated for 1 min. with chilled acetone to remove neutral lipids [15] and rehydrated with PBS. Cells were then incubated for 10 min. with 3.5 M guanidinium thiocyanate (GTC) in PBS. Cells were washed at least five times with PBS to quantitatively remove GTC before antibody labeling. The primary antibodies used were: anti-PrP antibody 5B2 (Santa Cruz, Sc-47730, mouse IgG1), neuron-specific anti-Tubb3 (R&D systems, clone TuJ-1, mouse IgG2a), and astrocyte-specific anti-GFAP (Dako, Z 0334, rabbit IgG). Fluorescence-conjugated, mouse subclass-specific secondary antibodies (AffiniPure) were purchased from Jackson Immunoresearch (Stratech, Ely, UK) and they are highly cross-adsorbed against rabbit proteins to avoid cross-reactivity with anti-rabbit antibodies. For immunostaining, primary and secondary antibodies were diluted in sterile-filtered Superblock (Pierce) solution/PBS (1:4; *v*/*v*), overnight at 4 °C. In some instances, cells were labeled for 10 min. with CellMask Deep Red plasma membrane stain (Thermo Fisher Scientific) prior to fixation, according to the manufacturer’s specifications. Images were captured with a Zeiss LSM 710 laser-scanning microscope and analyzed while using Zen imaging software (Carl Zeiss, Cambridge, UK).

### 2.9. Determination of Rod Length in Primary Neuronal Cultures

We used Volocity image analysis software (Perkin Elmer, Beaconsfield, UK) to measure the length of 5B2-positive aggregates in prion-infected primary neuronal cultures. Briefly, RML- and Me7-infected primary neuronal cultures from Prnp^−/−^ and Prnp^+/+^ FVB mice were fixed with 4% formalin and then washed with PBS. The cells were labelled with primary antibodies anti-GFAP and 5B2, followed by labeling with fluorescence-conjugated secondary antibodies. Serial z-stacks with a periodicity of 0.5 μm and a size of 5–10 μm were recorded and the size of 5B2-positive PrP aggregates were determined while using the Volocity line tool. The lengths of 5B2-positive PrP aggregates from 3–6 frames (60 μm × 60 μm) was determined and the size distribution of rods was calculated, including median values, first and third quartiles, and minimal/maximal lengths. The levels of significance were computed by non-parametric statistics (Kruskal-Wallis test with Dunn’s multiple comparisons).

## 3. Results

### 3.1. Isolation of Two Rare PK1 Cell Clones Potentially Permissive to Me7

While prion strains have classically been defined by their differences in disease incubation times, lesion profiles and electrophoretic mobilities, mounting evidence of strain-dependent differences in cell tropism [10,11,12,13,14] provides experimental approaches for examining the mechanistic underpinning of strain-dependent effects. The seminal observation that aggregates of the two mouse-adapted prion strains Me7 and 22L were associated with different cell types *in vivo* [13] suggests that cell models *in vitro* might inform about strain-dependent routes of uptake [14], trafficking and/or propagation. It is therefore critical to establish suitable cell models that may help to further our understanding in prion strain pathogenesis and selective neuronal vulnerability.

Mice succumb to prion disease with differences in disease incubation times following intracerebral inoculation with RML and Me7 [19], but some neuronal cell lines are differentially susceptible to these two prion strains. The mouse neuroblastoma N2a-derived clone PK1, for example, is highly susceptible to RML, but refractory to Me7 [12,16]. Notably, Me7 and RML were suggested to differ in their conformational stability following denaturation [20], but how such conformational constraints are associated with differences in prion propagation is unknown. We reasoned that isogenic cells that are differentially susceptible to either RML or Me7 may constitute a highly favorable model to study causes of “strain-ness”. Therefore, we challenged a large number of PK1 subclones with Me7 to assess whether it is possible to identify Me7-permissive PK1 clones. The Scrapie Cell assay (SCA), which is a quantitative cell-based infectivity assay [17], was used to determine the number of PrP^Sc^-infected cells. Out of 720 single cell clones that were incubated with Me7, two clones (PME1 and PME2) showed a low, but stable number of PrP^Sc^-positive cells at split six after incubation (Table 1). Of note, a prion-infected state after infection can only be maintained, if the rate of *de novo* prion formation exceeds the rates of cell doubling and degradative pathways, respectively [21]. We challenged sublines of PME1 and PME2 with three distinct mouse-adapted prion strains, Me7, RML, as well as 22L, and determined the distribution of infected cells to further evaluate whether the two identified subclones show broad susceptibility to mouse-adapted prion strains (Figure 1). While sublines of both cell clones yielded infected clones with > 600 PrP^Sc^-positive cells after infection with RML and 22L, respectively, only PME2 sublines were stably infected after challenge with Me7, which suggests that PME1 subclones might fail to maintain chronic infection (Figure 1).

### 3.2. Progressive Enrichment of Me7-Susceptible Cells by Subcloning

Therefore, we aimed to isolate highly Me7-susceptible PME2 cell clones by successive subcloning rounds. Notably, serial subcloning progressively enriched highly Me7-susceptible cell clones (Figure 2). Concomitantly with a > 20 fold enrichment of Me7-susceptible clones in each subcloning round (Figure 2B), the rate of prion propagation increased substantially (Figure 2C), resulting in highly Me7-susceptible cell clones in three subsequent rounds. To further characterize the enrichment of Me7-susceptible clones, we compared the changes in the number of PrP^Sc^-infected cells between three subsequent passages as a criterion of stable prion propagation, in accordance to the observation of acute and persistent prion infection [11]. While a progressive increase in the number of PrP^Sc^-infected cells during subsequent cell passages is indicative of persistent infection, a decrease might suggest that the prion propagation is not maintained over extended periods of time. As shown in Figure 2D, spot numbers from two subsequent cell passages were uncorrelated (R^2^ = 0.13) for second generation PME2 subclones, but highly correlated (R^2^ = 0.87) for third generation PME2-6D8 subclones, confirming the enrichment of Me7-susceptible cells at the population level. This result is in agreement with a shift of the population distribution towards highly Me7-infected cell clones in third generation clones, when compared to those of the second generation (Figure 2E). The most susceptible PME2 and PME2-D8 subclones are listed in Table 2, being ranked by susceptibility to Me7 propagation. In addition to highly Me7-susceptible cells, we also aimed to isolate the Me7-refractory cells from the same subclone, PME2-6D8. As explained earlier, such isogenic clones with diametrically opposite phenotypes might be highly valuable to identify strain-dependent cellular factors that are associated with infection. When we infected PME2-6D8 clones with Me7, RML, and 22L, respectively, we identified seven Me7-refractory clones (Table 3). Of note, Me7-refractory clones were still highly permissive to the prion strains 22L and RML, which argues that these clones did not shift into a general non-permissive state, but are selectively resistant to Me7 propagation. Therefore, these Me7-refractory PME2-6D8 clones are highly valuable to further identify cellular factors that are associated with Me7 infection. The enrichment of Me7-susceptible cells by successive rounds of single cell cloning also changed the odds of isolating 22L- and RML-refractory as well as susceptible cells. Notably, all of the clones tested in the final subcloning round were highly susceptible to 22L and RML, respectively (Appendix A). 

### 3.3. Me7-Infected PME2 Cells Deposit PrP^d^ Aggregates and Differ in Their PK Sensitivity from RML-Infected Cells

A representative highly Me7-susceptible clone, PME2-1H10, was plated out in chambered glass slides and cultured for six days to probe the presence of PrP^d^. The cells were fixed, treated with acetone and GTC, as specified in Materials and Methods, and then incubated with 5B2, an anti-PrP antibody that recognizes the N-terminal region of PrP [22]. Aggregates of PrP^d^ were readily detected in chronically infected, but not in uninfected PME2-1H10 cells (Figure 3A). Alterations in the biochemical characteristics of PrP^Sc^ upon transfer of prion strains from brain to cells have been reported previously [23,24]. To examine whether passage of Me7 in susceptible neuroblastoma PME2 cells alters the strain characteristics, we conducted biochemical strain typing by Western blotting. Note that, here, cell-adapted prion strains are denoted with a box bracket after the clone name, for instance PME2 [Me7], in line with the nomenclature by Li et al. [18] as explained in Materials and Methods. As shown in Figure 3B, passage of brain-adapted Me7 and RML in permissive PME2 cells led to considerable change in the electrophoretic mobility and glycosylation profile of PrP species. Whether the samples were digested with Proteinase K (PK), or left undigested, the strain types of PME2-adapted Me7 and RML looked alike, but greatly differed, when compared to brain-adapted Me7 and RML. This suggests that a single passage of Me7 and RML in susceptible PME2 cells gives rise to altered strain types and raises the question of whether propagation in PME2 cells renders the Me7 strain “RML-like”. As a first approximation, we investigated whether cell-adapted RML and Me7 differ in their PK-sensitivity, a criterion that informs regarding the differences in protein conformation of PrP species. We prepared cell homogenates of the infected PME2-6D8 cells by ribolyzation and subjected aliquots of 50 μg protein to a range of PK concentrations between zero and 60 μg/mL. As shown in Figure 3C,D, the PME2-6D8 [Me7] strain is considerably more PK-resistant than PME2-6D8 [RML], in keeping with the higher conformational stability of Me7 [20]. We harnessed cells that are discriminatory to the examined strains to further characterize cell and brain-adapted prion strains.

### 3.4. Evidence of Prion Strain Adaptation in Me7-Susceptible PME2 Clones

The cell panel assay is a rapid method for distinguishing prion strains [12] and it utilizes cells with known susceptibility to mouse-adapted prion strains. To further assess whether brain-adapted Me7 can be distinguished from the cell-adapted PME2-6D8 [Me7] strain, we used a panel of four distinct cell lines: Me7-refractory PK1 cells, Me7-suceptible (PME2-6D8), and -refractory (PME2-6D8-4H4, see Table 3) PME2 clones, as well as CAD5, a Me7-susceptible CNS catecholaminergic cell line [12]. All of the cells were incubated with 10^−4^ dilutions of brain homogenates (10%, *w*/*v*) or 5 × 10^−2^ dilutions of prion-infected cell homogenates and split 1:8 for five successive passages. Representative images in Figure 4A,B show PrP^Sc^-positive cells, detected as anti-PrP-immuno-positive black spots in wells of Elispot plates (Figure 4A,B) and quantified with an Elispot scanner as described previously [17]. As anticipated, the PME2-6D8 cells readily propagate both, brain and cell-adapted Me7 (Figure 4A,C). Infection of cells with RML and PME2-6D8 [RML] was used as a control and it shows a steady increase in the number of infected cells in subsequent passages for both strains (Figure 4C, right panel). Unexpectedly, PK1 cells that are refractory to Me7 readily propagated the PME2-6D8 [Me7] strain, which suggested that the strain properties of Me7 changed upon passage in PME2-6D8 cells. CAD5 cells were refractory to Me7 (Figure 4B,D), in disagreement to previously published work [12], suggesting that the cell line has lost susceptibility to Me7. The Me7-refractory clone PME2-6D8-4H4 (Table 3) failed to propagate Me7, but it readily propagated PME2-6D8 [Me7], which confirms the notion of prion strain adaptation (Figure 4B,E). In conclusion, we provide evidence that the prion strain Me7 adapts during cell passage in Me7-susceptible PME2 clones. A comparison between isogenic susceptible and refractory PME2-6D8 clones thus constitutes a compelling model to study strain adaptation.

### 3.5. Expanded Host Range of Cell-Adapted Me7 Prions in Primary Neuronal Cells

We cultured mixed primary neuronal cells from embryonic e17 FVB mouse brains for infection with brain- and cell-adapted strains to further investigate the phenomenon of prion strain adaptation and to expand the cell panel from immortal cell lines to a more complex system. Primary neuronal cultures have physiological properties that are remarkably similar to cortical neurons *in situ*, show frequent recurrent excitatory and inhibitory synaptic connections [25], and are a powerful tool for studying neurodegeneration.

We labeled primary neuronal cultures two weeks after infection with RML and Me7 using neuron, astrocyte, and PrP-specific antibodies to investigate whether the neuron-glial network recapitulates differences in prion strain discrimination. After fixation and pre-treatment according to Materials and Methods, we incubated cells with anti-Tubb3 (mouse IgG2a, neuron-specific), anti-GFAP (rabbit IgG, astrocyte-specific), and 5B2 (mouse IgG1, anti-PrP). We used fluorescence-conjugated, anti-mouse subclass-specific secondary antibodies (Affinipure, Jackson ImmunoResearch) that are highly cross-adsorbed against rabbit proteins to avoid cross-reactivity with anti-rabbit antibodies. In mixed cortico-hippocampal cultures, the degeneration of Tubb3-positive neurons is already apparent after two weeks of culture (Appendix A), which suggests that the contribution of neurons to prion propagation might be limited under the experimental conditions. As shown in Figure 5A, rod-like aggregates of PrP^d^ were detected in association with astrocytes at two weeks after incubation with RML, but not after incubation with Me7. This result was corroborated by infecting primary neuronal cultures from Prnp^−/−^ (ko) and Prnp^+/+^ (wt) mice with RML and Me7 (Figure 5B). Although PrP-immune-positive spots were occasionally detected in FVB ko mice at three weeks after infection with 10^−4^ RML, rod-like structures were frequently observed in FVB wt, but not in FVB ko mice. We determined the size distribution of rods in infected neuronal cultures from ko and wt mice while using Volocity image analysis software (Figure 5C). A significant increase in rod-like PrP structures with lengths of up to 17 microns were detected in neuronal cultures from wt mice at three weeks after infection with RML, while 5B2-positive puncta in cultures of ko mice ranged from 0.5 to 2 microns. The size distribution of 5B2-positive puncta in Me7-infected neuronal cultures from FVB wt mice did not significantly differ from those of replication-incompetent ko mice. The detection of rod-like PrP^d^ aggregates with anti-PrP antibody 5B2 that recognizes an epitope at the PrP N-terminus (48–50) [26] is in agreement with previous work [27], where PrP “strings” were detected in infected cell lines and in the hippocampus of FVB mice with 8B4, an anti-PrP antibody that also recognizes the N-terminus of PrP (37–44) [28].

We next tested susceptibility to PME2-adapted Me7 since mixed primary neuronal cultures were refractory to Me7 under the culture conditions used. Of note, rod-like PrP^d^ aggregates were detected in association with astrocytes following infection with PME2-adapted Me7 (Figure 5D) and about half of the astrocyte population in primary neuronal cultures showed PrP^d^ aggregates (Figure 5E). Astrocytes with PrP^d^ aggregates were also confirmed after infection with PME2-adapted RML, but were absent following incubation with Me7 and uninfected CD1 brain homogenates, respectively (Figure 5D,E). In addition to phenomenological evidence of rod-like PrP^d^ aggregates in Figure 5, we conducted cell-based infectivity assays while using the PME2-6D8 clone to determine the prion titers of primary neuronal cultures after infection with brain and cell homogenates (Appendix A). Briefly, we infected primary neuronal cultures from FVB ko and wt mice with RML, Me7, PME2-6D8 [Me7], and uninfected CD1. Two and three weeks after infection, we ribolyzed the neuronal cultures and infected PME2-6D8 cells with serially 1:10 diluted homogenates as described in Materials and Methods. While infectious titers of RML-infected primary neuronal cultures from FVB wt mice exceeded those of the FVB ko mice by one order of magnitude (5.6 versus 4.6 log [TCIUs]) at three weeks after infection, infectious titers of residual inoculum in neuronal cultures from FVB ko mice did not decrease in a time-dependent manner, which suggested that prions are not degraded in mixed neuronal cultures under these experimental conditions. Given these limitations, the rod-like PrP^d^ phenotype that was observed in primary neuronal cultures from wt, but not in those from ko mice (Figure 5B) unambiguously demonstrates that prions are replicated in astrocytes.

In conclusion, our results confirm that passage of Me7 in permissive PME2 cells generates an altered prion strain with expanded host range. Furthermore, we show that the infected astrocytes are decorated with anti-PrP immuno-positive PrP^d^ rods.

## 4. Discussion

We derived highly Me7-susceptible cell clones from a Me7-refractory mouse neuroblastoma cell line, PK1, by single cell cloning, a study that shows unexpected plasticity of neuroblastoma cells to mouse-adapted prion strains. We further provide evidence that the strain properties of Me7 are altered by passage in susceptible clones, which results in a novel prion strain with expanded host range, when compared to the original mouse-adapted strain. This notion was confirmed by the high infection rate of Me7-refractory astrocytes in primary neuronal cultures by cell-adapted Me7. The detection of 5B2-immuno-positive micrometer-long rods of PrP^d^ in association with astrocytes indicates a yet undefined mechanism of seeded prion aggregation. In conclusion, we established a novel panel of isogenic cell clones that can be harnessed as a cellular model for prion strain selection and adaptation, and that might provide guiding principles to understand the causes for selective neuronal vulnerability.

### 4.1. A Bottom-Up Approach to Study Selective Neuronal Vulnerability Using Strain-Selective Isogenic Neuronal Cells

In neurodegenerative diseases, neuronal populations of distinct brain areas degenerate, a phenomenon that is known as selective neuronal vulnerability. While amyotrophic lateral sclerosis is associated with selective degeneration of motor neurons, disease progression in Parkinson’s disease leads to the degeneration of dopaminergic neurons in the substantia nigra. In prion diseases, selective neuronal vulnerability is linked to the degeneration of particular brain areas in a strain-dependent manner.

Considerable experimental evidence suggests that the lesion profiles of microvacuolation (spongiform degeneration) and PrP^Sc^ distribution are associated with distinct prion strains (for reviews see [29,30,31]). Developed originally in mouse models of scrapie [32], lesion profiling has been applied to different prion diseases, like BSE [33], Scrapie [34], and CJD [35], to characterize prion neuropathology at preclinical and clinical stages of prion diseases. While lesion profiling is adding valuable empirical data to discriminate prion strains, it does not inherently provide an experimental approach to examine the molecular underpinnings of selective neuronal vulnerability. In two recent reports, strain-dependent differences in cell tropism have been reported for sheep scrapie [36], and experimental scrapie in mice [13]. Me7 was mainly detected in association with neurons and neuropil, while 22L was associated with microglia [13]. In keeping with the notion of strain-dependent cell tropism, many immortalized cell lines are distinct in their susceptibility to prion strains [37,38,39], a propensity that is harnessed in the cell panel assay to distinguish prion strains [12]. This raises the question as to whether strain-dependent differences in cell tropism, as observed *in vivo* by immunohistochemistry [13,36], can be recapitulated *in vitro* and how such studies have to be designed to make inferences on selective prion strain uptake and/or propagation pathways.

Although a variety of cell models with susceptibility to more than fourteen mouse-adapted strains and natural TSE isolates exist [39], cellular factors that account for the cell tropism of strains are unknown. A direct comparison of the transcriptional profiles of cells with distinct susceptibility to prion strains is hampered by considerable differences in the gene expression profiles of cell types from different tissues or cancers [40]. Such “noise” at the transcriptome level may greatly limit the identification of gene expression profiles that are associated with a relevant phenotype, like susceptibility to prion propagation. The recent identification of a gene signature that is associated with susceptibility to prion replication was harnessed by isolation of isogenic resistant subclones of highly prion-susceptible cells [15]. Single cell cloning is an excellent way to limit genetic variation, while harnessing phenotypic heterogeneity to establish favorable cell models. Phenotypic heterogeneity of clonally derived cells is often associated with epigenetic variation [41] and triggered by stochastic gene activation events [42], which is an important contributor to reprogramming transitions [43,44]. Systematic studies into stochastic gene expression events have been facilitated by single cell transcriptomics (for recent reviews see [44,45,46]). Conceptually, identification of differentially expressed genes in cognate cells with distinct susceptibility to prion strains is complex and requires complementary sets of susceptible and refractory cells, preferably with exclusive susceptibility to distinct prion strains. In this pilot study, we enriched for Me7-susceptible subclones from Me7-refractory PK1 cells and monitored susceptibility of the progeny to RML and 22L to examine whether PK1-derived sublines are they a sufficient model to investigate the causes of strain-dependent differences in cell tropism.

In this study, we isolated PK1-derived clonal lines with an expanded host range to mouse-adapted prion strains. Single cell cloning led to a 600-fold enrichment of Me7-susceptible clones (Figure 2). Figure 6 depicts the percentage of cell clones that are susceptible to Me7, RML, and 22L during a three-stage subcloning procedure. Notably, the enrichment of Me7-susceptible clones is accompanied by a concomitant increase in susceptibility to 22L (Figure 6). While an increase in the proportion of Me7-susceptible cells is highly favorable, a shift in susceptibility for other strains may limit the odds of isolating critical phenotypic clones. For instance, cells with a phenotype “Me7-susceptible” (Me7sus), “22L-refractory” (22L_ref_), in short Me7_sus_ 22L_ref_, were isolated with a likelihood of 7.8 × 10^−3^ in the second round of subcloning, while no cells of this phenotype could be isolated in the third round due to the lack of cells with a 22L_ref_ phenotype (Appendix A and Figure 6). Such data on the likelihood of isolating distinct strain-permissive cells in this pilot experiment is pivotal for the design of gene expression analyses. For instance, to identify genes that are associated with susceptibility to Me7, but not to 22L or vice versa, a panel of cells with phenotypes Me7_sus_ 22L_ref_, Me7_ref_ 22L_sus_, Me7_sus_ 22L_sus_, Me7_ref_, and 22L_ref_ are required and they can be derived from PK1, PME2, and PME2-6D8 according to the most likely parental cell lines according to Figure 6.

### 4.2. Prion Strain Variation and Adaptation

A multitude of different prion strains has been isolated from natural prion diseases in animals and humans and further propagated in inbred mice. Prion strains were proposed to consist of an ensemble of distinct molecules and propagation in a host may lead to amplification of a dominant PrP^Sc^ type [47]. Prion strains have been largely defined by virtue of their neuropathological and biochemical characteristics. “Strain typing”, the most widely used method for classifying prion strains, relies on protease digestion of isolates, followed by electrophoretic separation, a procedure that has been used to classify human prion strains [48,49]. Strain typing may be of limited use for atypical, protease-sensitive strains [50,51], and prion strain mixtures [52].

That prion strains may change their properties upon transfer into a different host has been demonstrated *in vivo* and *in vitro*. Intracerebral inoculation of BSE prions into two inbred mouse strains, SJL and C57BL/6, resulted in propagation of two distinct prion strains, as confirmed by differences in disease incubation times, neuropathology, and biochemical properties [53]. An atypical BSE strain acquired strain properties, typically found in the epidemic BSE agent upon transmission into ovine PrP transgenic mice [54]. The propagation of the mouse-adapted prion strain 22L in PK1 cells led to progressive changes in cell tropism and strain properties of the cell-adapted prion strain [18].

That prion strains gradually and reversibly adapt to a new host environment has been carefully studied with the cell panel assay by virtue of drug selection with small molecule inhibitors, like swainsonine, an inhibitor of Golgi alpha-mannosidase II [18]. When propagated in PK1 cells in the presence of swainsonine, 22L gradually acquired swainsonine resistance, but then reverted back to its original strain properties after passage in mouse brains. In similar experiments with swainsonine-sensitive prions, the rise of swainsonine-resistant prions suggested that prion assemblies adapt to drug-mediated selection pressure [18].

Our study contributes additional insights into the biology of strain adaptation and provides a cell model to investigate the underlying biological pathways of this phenomenon. Notably, our evidence that Me7-refractory cells propagate cell-adapted Me7 (Figure 4) suggests that strain adaptation is linked to prion propagation in a permissive host, which gives rise to a strain variant with an expanded host range. Given that Me7-refractory cells (PK1, PME2-6D8-4H4) lack a factor that is required for Me7 propagation, strain adaptation may be mediated by PME2-dependent alterations in strain properties. Post-translational modifications, like glycosylation [55,56,57] and sialylation [58], have been suggested to account for the conformational diversity of prion strains and cell-based systems could be instrumental to provide clues about the elusive link between biological and conformational variation of prion strains.

## Figures and Tables

**Figure 1 viruses-11-00888-f001:**
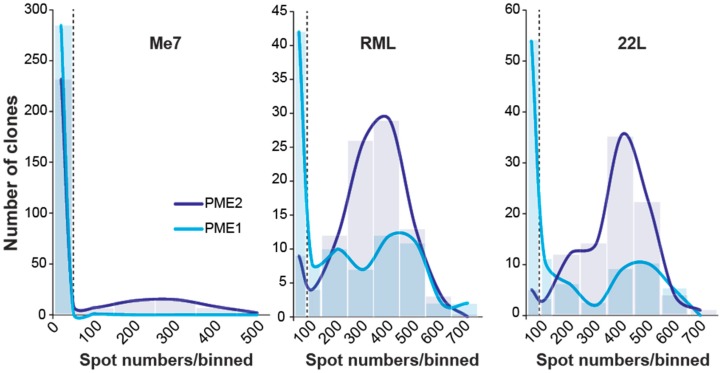
Distribution of prion-infected PME1 and PME2 sublines following infection with Me7, RML and 22L. Cultures of PME1 and PME2 cells were challenged with a 10^−3^ dilution of mouse-adapted prion strain homogenates Me7, RML and 22L (10% *w*/*v*). Cells were split 1:8 twice a week to dilute out the inoculum. After seven passages, limiting dilutions of cells were plated and single cell clones isolated to determine the state of prion infection using the Scrapie Cell assay. A total of 288 Me7-infected clones and 96 RML- and 22L-infected clones were isolated for each of the parental clones PME1 and PME2. To plot the distribution of infected cells per homogenate and cell line, the number of infected cells (spots) were binned into groups of 100 and plotted against the corresponding number of clones; the first two bins contain 0–50 and 50–100 spots. The dashed line at 50 spots denotes a cut-off value for scoring infected cells over background.

**Figure 2 viruses-11-00888-f002:**
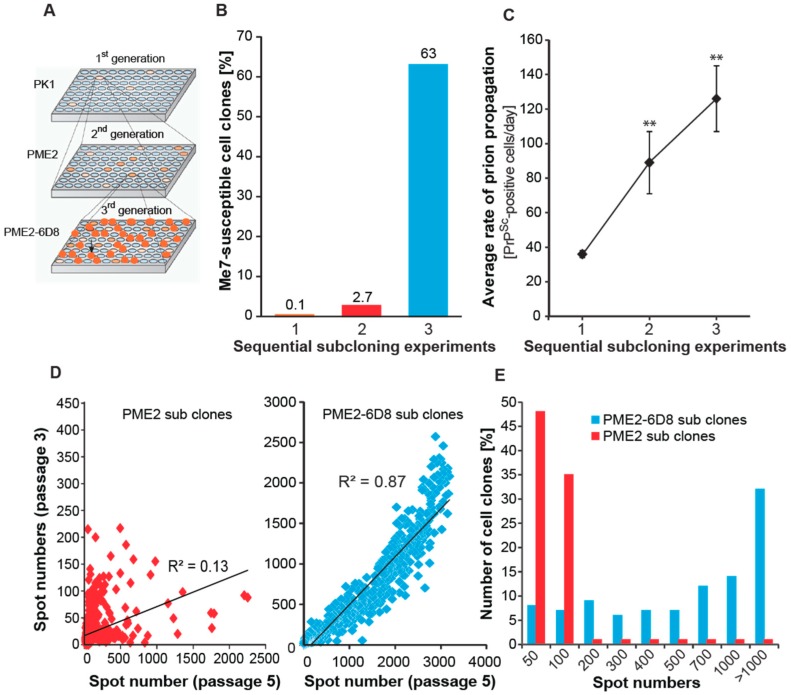
Progressive enrichment of Me7-susceptible cells by single cell cloning. (**A**) Schematic representation of a serial subcloning strategy to identify Me7-susceptible cell clones. Plates of single PK1 cell clones were infected with Me7, while duplicate plates remained uninfected (not shown). The susceptible clone PME2 was expanded at limiting dilution to isolate 2nd generation clones (PME2 progeny). This protocol was repeated to isolate 3rd generation (PME2-6D8) clones. (**B**) Progressive enrichment of Me7-susceptible cell clones. Data represents the percentage of Me7-positive cell clones in three subsequent subcloning experiments according to **A**. The total number of clones characterized in experiments 1, 2 and 3 were 720, 960, and 468, respectively. (**C**) The rate of prion propagation in prion-infected cells increases progressively upon sequential subcloning. Data represents average rates of prion propagation for five representative cell clones and is expressed as number of PrPSc-positive cells per day ± SD. Statistical significance was calculated using Student’s t-test (** *p* < 0.001) (**D**) Enrichment of highly Me7-susceptible cells in 3^rd^ generation (PME2-6D8) subclones. Data represents spot numbers of individual clones from two cell passages (passage 3 and 5). An increase in spot number is indicative of perpetuating prion replication. Correlation coefficients R^2^ for linear regression analysis is shown. (**E**) Distribution of Me7-susceptible PME2 and PME2-6D8 clones. Cell clones were challenged with a 10^−5^ dilution of Me7 brain homogenate (10% *w*/*v*). After five cell passages, the number of spots was determined by Scrapie Cell assay.

**Figure 3 viruses-11-00888-f003:**
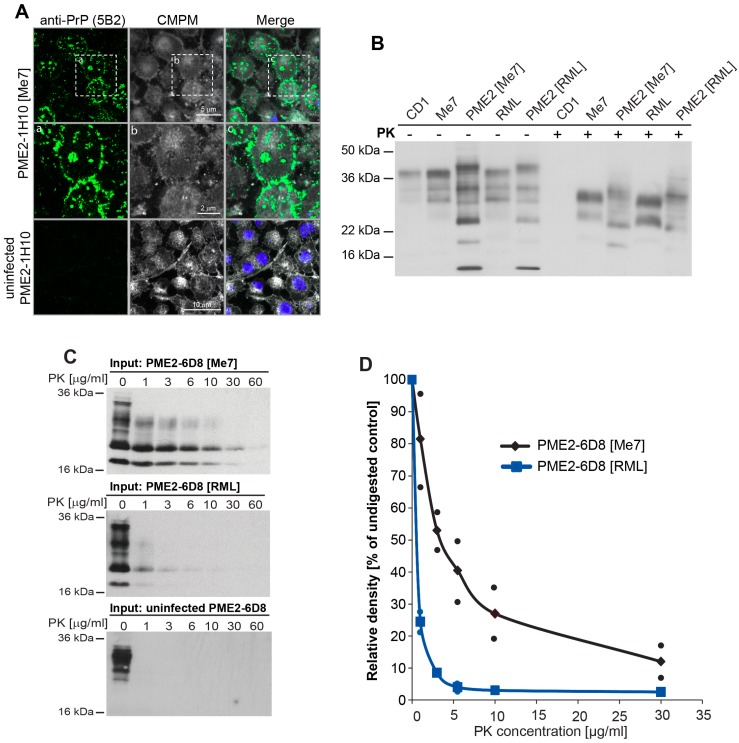
Chronically Me7 infected PME2 cells deposit PrP^d^ aggregates and differ in their PK sensitivity from RML-infected cells. (**A**) Laser-scanning microscopy images of uninfected and chronically Me7-infected PME2 subclones. Eleven passages after infection of PME2-1H10 cells with a 10^−5^ dilution of Me7 brain homogenate, cells were plated out in chamber slides and cultured for 6 days. Cells were treated post fixation according to Materials and Methods and labeled with 5B2 and CellMask Deep Red plasma membrane (CMPM) stain (Thermo Scientific) according to the manufacturer’s specification. Magnifications of infected PME2-1H10 cells are shown in the panel below as indicated. (**B**) Western blot of brain homogenates CD1 (uninfected), Me7 and RML and cell-adapted prion strains PME2 [Me7] and PME2 [RML] with and without Proteinase K (PK) digestion. (**C**,**D**) Pools of three Me7- and RML-infected PME-6D8 clones were ribolyzed and protein concentrations were determined by bicinchoninic acid (BCA). Aliquots of 50 μg cell homogenate were digested with Proteinase K (PK) at the concentration range specified. Representative Western blot are shown in C and PK-sensitivity of PrP^Sc^ quantified (D) as specified in Materials and Methods.

**Figure 4 viruses-11-00888-f004:**
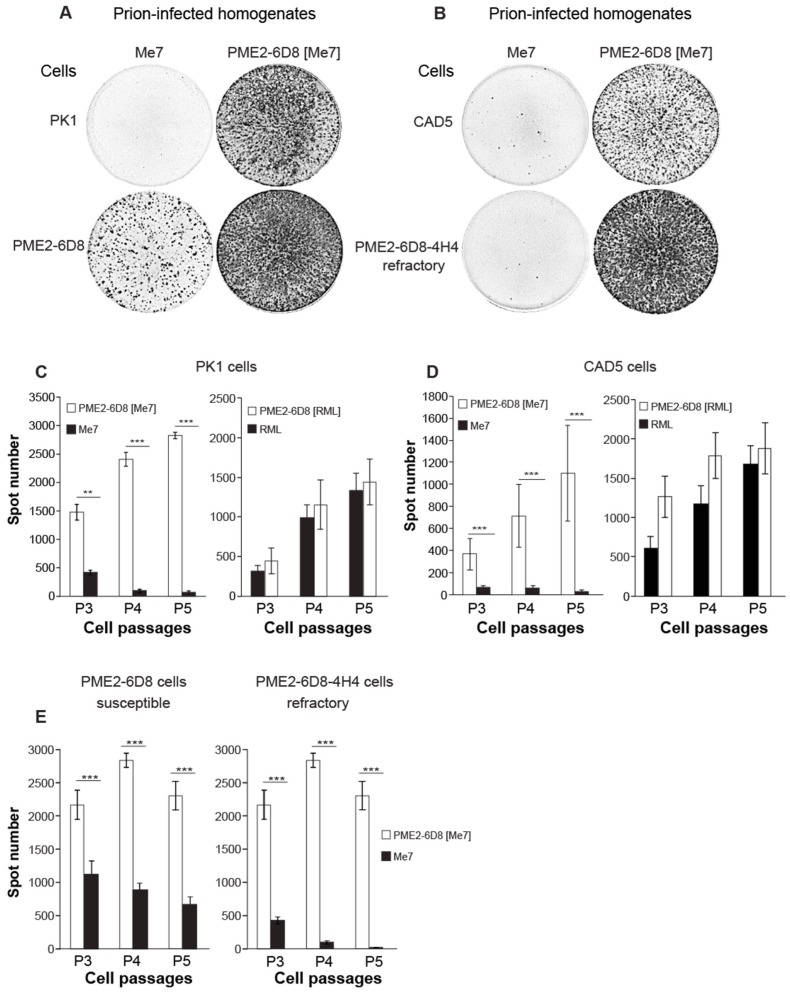
Passage of Me7 in susceptible PME2 clones leads to prion strain adaptation. (**A**–**E**) To determine the cell tropism of brain-adapted and cell-adapted prions, Me7-susceptible (PME2-6D8) and -refractory (PK1, CAD5, and PME2-6D8-4H4) cells were infected with 10^−4^ dilutions of Me7 and RML brain homogenates and with 5 × 10^−2^ dilutions of PME2-6D8 [Me7] and PME2-6D8 [RML] cell homogenates, respectively. After five cell passages, aliquots of 2,000 cells were transferred to Elispot plates and analyzed by SCA according to Materials and Methods. (**A**,**B**) Representative wells from Elispot plates with prion-infected (black spots) and refractory (blank wells) cells, immuno-labeled with anti-PrP antibody ICSM18 and secondary alkaline phosphatase-conjugated antibody, according to Materials and Methods. (**C**–**E**) Data analysis of SCAs. Data represents mean spot numbers ± SD for a total of twelve wells per cell line and infection as specified over three subsequent cell passages. Statistical significance was calculated while using Student’s t-test for two significance levels (** *p* < 0.001, *** *p* < 0.0001).

**Figure 5 viruses-11-00888-f005:**
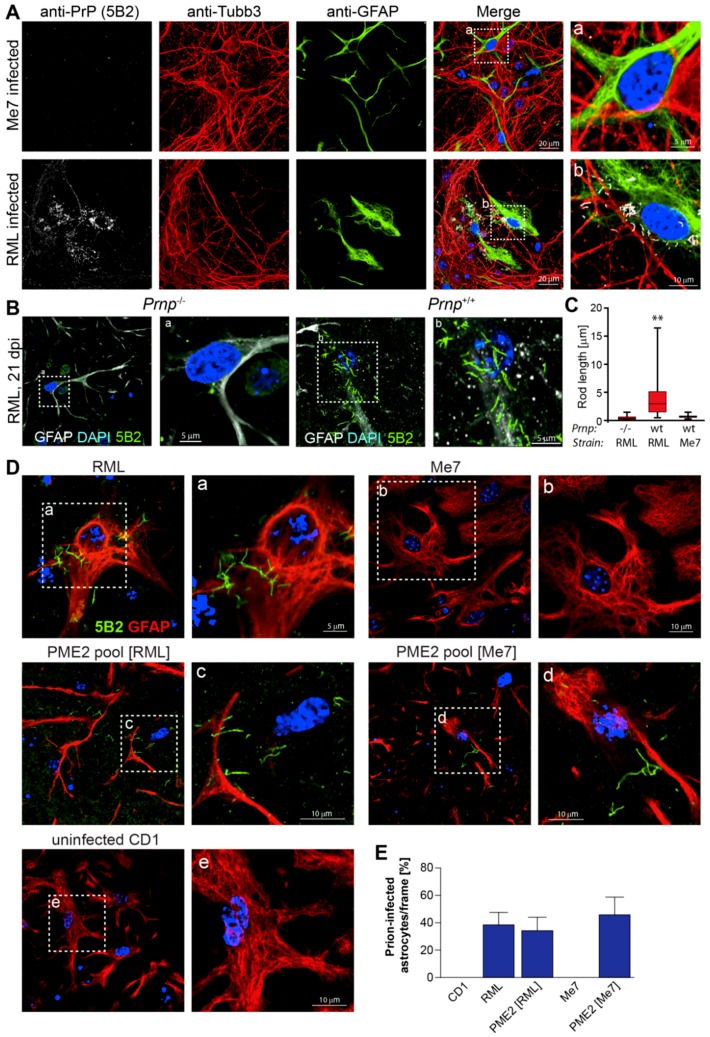
Expanded host range of cell-adapted Me7 prions in primary neuronal cells. Representative laser-scanning microscopy images of primary neuronal cells three weeks after infection with brain- and cell-adapted prion strains. Primary cortico-hippocampal cultures were prepared from embryonic e17 FVB mouse brains as specified in Materials and Methods. (**A**) Primary neuronal cultures were plated into chambered cover glass slides, coated with poly-L lysine. After two days of culture, primary cells were infected with 10^−4^ dilutions of Me7 (10% *w*/*v*) and RML (10% *w*/*v*), respectively. After 2 weeks of culture, cells were fixed for 15 min. with 3.7 % formalin and treated with acetone and GTC as described in Materials and Methods. Fixed cells were incubated with primary antibodies anti-PrP 5B2 (mouse IgG1; 1:500), anti-Tubb3 (mouse IgG2a; 1:2000) and anti-GFAP (rabbit IgG; 1:2000), followed by highly cross-adsorbed Jackson secondary antibodies (Affinipure) as described in Materials and Methods. Rod-like PrP^d^ aggregates were detected in RML-, but not in Me7-infected cells. The hatched insert denotes magnified areas displayed. (**B**,**C**) Primary cortico-hippocampal cultures from FVB Prnp^−/−^ and Prnp^+/+^ (wt) mice were infected with 10^−4^ dilutions of RML (**B**,**C**) and Me7 (**C**) (**D**) and fixed at three weeks post infection. Representative confocal images of neuronal cultures, double labelled with anti-GFAP and 5B2 are shown in (**B**). The length distribution of rod-like PrP^d^ aggregates was determined using Volocity as described in Materials and Methods and are depicted as box plots in (**C**). ** *p* < 0.001 (Kruskal-Wallis test). Primary neuronal cells were infected with prion strains RML, Me7, uninfected CD1 and cell-adapted prion strains PME2 pool [RML] and PME2 pool [Me7], respectively. Cell culture and labeling protocols were conducted three weeks after infection and are essentially as described above. All of the cells shown were double-labeled with 5B2 and anti-GFAP. (**E**) Quantitative analysis of the number of PrP^d^-bearing astrocytes in (**D**). The number of infected astrocytes was determined in twenty randomly selected frames with an area of 17,000 μm^2^ each. Data represent mean values ± SEM.

**Figure 6 viruses-11-00888-f006:**
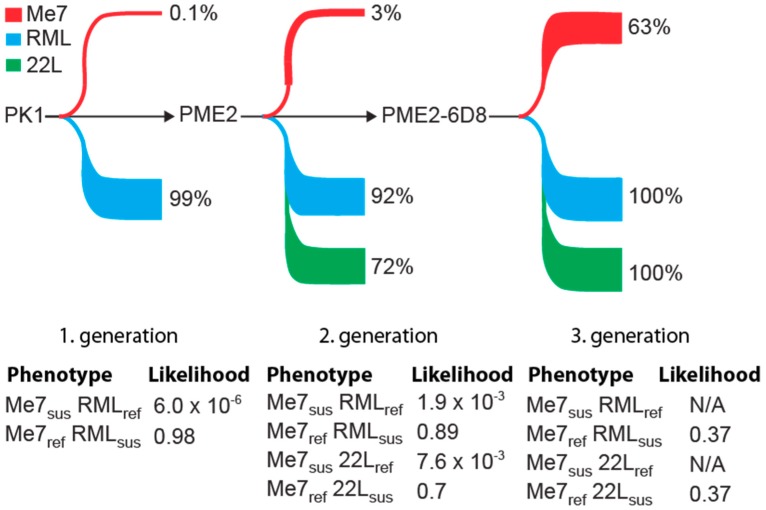
Phenotypic changes in susceptibility to mouse-adapted prion strains during enrichment of Me7-susceptible cells. Schematic representation of changes in susceptibility of PK1-derived clones to the three mouse adapted prion strains Me7, RML, and 22L during enrichment of a Me7-susceptible phenotype. Depicted is the proportion of cells susceptible to the specified prion strains. Likelihoods of isolating distinct phenotypic cells were calculated from data of Appendix A. N/A: not applicable; briefly, phenotypes RML- and 22L- are not available in third generation cells, since all the clones are permissive to RML and 22L, respectively.

**Table 1 viruses-11-00888-t001:** Rare PK1 subclones are weakly permissive to Me7 prions.

PK1 Subclones	Spot Number
PME2	378
PME1	351
16 clones	100–200
42 clones	50–100
660 clones	0–50

A total of 720 single PK1 subclones were isolated and challenged with a 2 × 10^−6^ dilution of Me7 brain homogenate (10% *w*/*v*). After six passages, the number of PrP^Sc^-positive cells (spot number) was determined by SCA as specified in Materials and Methods. The cut-off value for scoring prion-infected cells was set to 50 spots to account for false-positive background spots in control wells.

**Table 2 viruses-11-00888-t002:** Susceptibility of PME2 and PME2-6D8 clones to Me7.

	**Scrapie Cell Assay**	
**PME2 Subclones (Round 1)**	**PrP^Sc^-Infected Cells (Passage 3)**	**PrP^Sc^-Infected Cells (Passage 5)**
*1H10*	88	2235
*1F5*	92	2206
*1B2*	59	1748
*3B3*	58	1788
*6D8*	98	1356
*2G6*	49	1229
*2E9*	20	1287
*1A8*	7	10
*5F12*	13	11
**PME2-6D8 Subclones (Round 2)**	**PrP^Sc^-Infected Cells (Passage 3)**	**PrP^Sc^-Infected Cells (Passage 5)**
*4F11*	2079	3191
*2C6*	1863	3169
*5B9*	2178	3152
*5B1*	1618	3147
*5F1*	2131	3142
*1G1*	1844	3121
*5E1*	1997	3103
*5C2*	2030	3071
*PME2*	41 ± 8	137 ± 42

Subclones of PME2 and PME2-6D8, a PME2 progenitor isolated during the 2^nd^ subcloning experiment (Figure 2A,B) were challenged with a 10^−5^ dilution of Me7 brain homogenate. Three days after infection, cells were split 1:8 into OFCS and grown for 3–4 days. After two further cell passages, cells were counted and an aliquot of 85 μL transferred onto ELISPOT plates to determine the number of PrP^Sc^-infected cells as specified in Materials and Methods. Two further passages were conducted for passage five readings. To compare the number of PrP^Sc^-infected cells for all subclones with the parental clone PME2, said cell clone was infected with Me7 in parallel and processed under the same conditions. Results for PME2 represent average values of 12 wells ± SEM. All other results represent single reads due to the high number of clones tested.

**Table 3 viruses-11-00888-t003:** Differential susceptibilities of Me7-refractory and Me7-susceptible PME2-6D8 cell clones to three mouse-adapted brain homogenates.

	**Number of PrP^Sc^-Infected Cells after Infection with**		
**Me7-Refractory PME2-6D8 Subclones**	**Me7**	**22L**	**RML**
5E6	3	3038	2881
4C7	4	3151	2910
4D4	5	3170	2937
4C12	6	2786	2724
5G5	6	2949	2851
5E2	7	2935	2656
4H4	8	2723	2675
**Me7-Susceptible PME2-6D8 Subclones**	**Me7**	**22L**	**RML**
4F11	3191	2849	2883
2C6	3169	2858	2763
5B9	3152	2898	2824
5B1	3147	2772	3064
5F1	3142	2420	2368
1G1	3121	2602	2293
5E1	3103	2576	2840

The specified Me7-refractory and -susceptible PME2-6D8 clones were challenged with a 10^−5^ dilution of Me7, RML, and 22L brain homogenates. Data represent the number of PrP^Sc^-positive cells at passage 5 following infection, as determined by Scrapie Cell assay (SCA), as specified in Materials and Methods.

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
