# Peer review of "A New Cell Model for Investigating Prion Strain Selection and Adaptation"

_viruses, 2019, doi:10.3390/v11100888_

Round 1
Reviewer 1 Report
In the revised version of the manuscript, the authors have addressed all my concerns satisfactorily.
From my experience, the time-dependent decrease of prion infectivity in primary neurons from PrP knockout mice is very low. By comparison that in PrP convertible cultures is increasing significantly over time. This joins what the authors observed with their scrapie cell assay. I thus trust that the authors have succeeded in infecting their primary cultured neurons with RML.
Reviewer 2 Report
Questions and concerns have been adequately addressed. The difference in migration between ME7 and the PME adapted ME7 is convincing that PrP is replicating in these cells
This manuscript is a resubmission of an earlier submission. The following is a list of the peer review reports and author responses from that submission.
Round 1
Reviewer 1 Report
N2a mouse neuroblastoma cells are a convenient model to study the cell biology of prions. These cells and the derived subclones show differential susceptibility to mouse prion strains. In particular, N2a cells are permissive to RML prions and refractory to ME7 prions.
Here, the authors report on the selection of N2a subclones progressively permissive to ME7 prions. These prions show differential behavior in the so-called cell panel assay, suggesting that ME7 prions have ‘mutated’ on adaptation to the N2a subclones.
This part of the work is a nice and justified study. The new cell clones identified may help to better understand prion cell/tissue tropism (a true black box).
The authors also investigate whether primary cultured neurons or astrocytes from wild-type mice are permissive to RML, Me7 or N2a-adapted ME7 prions. This part of the work was much less convincing and the added value to the overall work less evident, mostly because key controls and key experiments are missing:
- Parallel exposure of primary cultured neurons and astrocytes from PrP knock-out mice should be performed to demonstrate that neosynthetized PrPd and not residual inoculum is detected at 14 days post-infection (particularly in figure 5B). Alternatively, or in addition, authors may show a time-course accumulation of prions.
- Orthogonal methods should be used to demonstrate presence or absence of PrPd accumulation in the cultures:
o In panel 5A, it sounds that neurons exposed to ME7 prions may accumulate low levels of PrPd.
o In Figure 5B, a and c panels, the pattern of PrPd accumulation sounds weird: usually, PrPd detection on IF look punctiform, and reminiscent of PrPd aggregate deposition (as in figure 5A). In these panels, most of the signal is diffuse, not cell-associated and the strongest signals look atypical.
o PrPd electrophoric pattern in these cultures should be compared to the pattern in N2a cells, to examine whether the 'mutated' ME7 profile from N2a subclones is conserved in astrocytes.
Overall, I wonder whether the primary cultures are truly infected.
Author Response
"Please see the attachment."

Reviewer 2 Report
Development of cell models for investigation of strain selection would be exceedingly helpful in the prion field. The selection of new strains must, however, be validated by all the criteria needed to identify new strains in animal models to demonstrate that the selection is analogous to in vivo adaptation and selection.
1) The cell line developed has unique properties--initially making the cell line susceptible to ME7. It appears, however, that the cell line becomes more refractory to ME7 upon further passage (retaining susceptibility to the cell-adapted ME7; Figure 4E). Can the authors comment on this? Does this not affect the utility of the cell line for parsing out the cellular factors involved in strain selection?
2) The cell-adapted ME7, on Western blot, looks very similar to RML agent (Figure 3B). The samples are not run in adjacent lanes on the Western blot---doing so would clarify the similarity/difference in the glycoform profiles.
3) One possible interpretation of the data is that the initial ME7 inocula contained a very small amount of RML agent---that was amplified through subsequent passage. Based on my experiences with in vivo adaptation, this is always a difficult interpretation to rule out. Is there any evidence for or against this interpretation?
4) To clearly demonstrate strain selection and development of an expanded host range, it is important to compare all strains (ME7, RML as well as the cell-adapted ME7 and RML) in mice.
Author Response
"Please see the attachment."
